

# Responses of fungal communities at different soil depths to grazing intensity in a desert steppe

Xiangjian Tu[1,2], Paul C. Struik[3], Shixian Sun[2], Zhang Wenbo[2], Yong Zhang[2], Ke Jin[2] and Zhen Wang[2,4]

[1] Key Laboratory of National Forestry and Grassland Administration on Grassland Resources and Ecology in the Yellow River Deltar, College of Grassland Science, Qingdao Agricultural University, Qingdao, Shandong, China
[2] Institute of Grassland Research, Chinese Academy of Agricultural Sciences, Hohhot, Inner Mongolia, China
[3] Department of Plant Sciences, Centre for Crop Systems Analysis, Wageningen University and Research, Wageningen, Provincie Gelderland, Netherlands
[4] Key Laboratory of Grassland Ecology and Restoration, Ministry of Agriculture, Hohhot, Inner Mongolia, China

Corresponding authors
Ke Jin, jinke@caas.cn
Zhen Wang, wangzhen0318@126.com

## ABSTRACT

Grazing can alter the physicochemical properties of soil and quickly influence the composition of microbial communities. However, the effects of grazing intensity on fungal community composition in different soil depth remain unclear. On the Inner Mongolia Plateau, we studied the effects of grazing intensity treatments including no grazing (NG), light grazing (LG), moderate grazing (MG), heavy grazing (HG), and over grazing (OG) on the physicochemical properties and fungal community composition of surface (0–20 cm) and subsurface (20–40 cm) soil layers. The $\alpha$-diversity of fungi in subsurface soil decreased under the influence of grazing. The relative abundance of Ascomycota in the subsoil was higher than that in the topsoil, while the situation of Basidiomycota was the opposite. This was caused by the differences in the soil carbon (C) environment for the growth of oligotrophic and copiotrophic fungi. In the subsoil, grazing affected nutrient contents such as soil organic matter (SOM) and total nitrogen (TN), resulting in significantly lower relative abundance of Ortierellomycota under LG, HG, and OG than in the NG. HG showed much higher relative abundance of Glomeromycota. Results of a multiple regression tree (MRT) analysis revealed that TN and nitrate nitrogen affected the fungal $\alpha$-diversity in top- and subsoils, respectively; the main driving factor regulating fungal community changes was soil water content (SWC) in the topsoil, while it was ammonium nitrogen and nitrate nitrogen in the subsoil. The results of our study indicate that grazing changes the soil environment by changing TN, SWC, nitrate nitrogen, ammonium nitrogen, and affects the diversity and community structure of soil fungi. This provides empirical support for coping with the impact of grazing on soil microbiomes in desert steppes.

## INTRODUCTION

Grazing is the most common land use strategy for grasslands. It can alter soil environments and the diversity and composition of soil microorganisms through livestock feeding, trampling, and the excretion of manure and urine (*Romero-Ruiz et al., 2023*; *Diao et al., 2024*). Herbivores can alter the quantity and quality of organic matter such as plant litter and manure excretion, which influence the input of soil carbon (C) and nitrogen (N), and subsequently affect soil microbial community composition (*Yao et al., 2023*). Livestock trampling, a component of grazing management, can increase soil compaction and reduce water holding capacity, while also increasing the diversity of soil microorganisms (*Treweek et al., 2016*). The impact of grazing intensity on soil physicochemical properties (such as soil organic matter, total nitrogen, *etc.*) is not linear (*Dai et al., 2022*). Appropriate grazing intensity can increase soil fertility, enzyme activity, and microbial biomass (*Serrano et al., 2024*; *Wan et al., 2021*). However, improper grazing intensity, especially over grazing, can limit the ecological functions and services of grasslands, threatening the balance between livestock grazing, environmental resources, and ecosystem stability (*Lan et al., 2023*). Currently, over grazing is widely recognized as the main cause of degradation in arid and semi-arid grasslands globally (*Wang et al., 2023*). Soil fungi (such as arbuscular mycorrhizal fungi) play a crucial role in the restoration process of degraded grasslands (*Li et al., 2023a*; *Li et al., 2023b*; *Zhou et al., 2024*). However, there is limited understanding of the relationship between grazing intensity and the diversity and community composition of soil fungi.

Soil microorganisms, as key players in the material cycle and energy flow, actively regulate ecological processes. An increase in their diversity can enhance the multifunctionality of ecosystems, such as litter decomposition and nutrient cycling (*Khatri-Chhetri et al., 2023*). The impact of grazing on soil physicochemical properties can be mitigated by certain soil microorganisms. For example, grazing activities can lead to a reduction in soil organic carbon, whereas microorganisms can enhance carbon metabolic pathways by increasing their element ratio thresholds and biomass, thereby increasing the amount of organic carbon derived from microbial sources (*Zhang et al., 2024*). Compared to bacteria, fungi can more effectively store and assimilate nutrients, and they are more closely associated with ecosystem multifunctionality (*De Vries et al., 2006*; *Li et al., 2022*). In recent years, there has been increasing interest in the diversity of fungi related to grazing intensity and fungi impact on various ecosystem functions (*Wang et al., 2021*; *Zhang et al., 2021*). However, most researchers have focused their attention on the effects of grazing on grassland plant diversity, physicochemical properties of the topsoil, and microbial diversity and function of the topsoil (*Yu et al., 2019*; *Du et al., 2023*; *Zhang et al., 2023a*; *Zhang et al., 2023b*). The effect of grazing on soil fungi in the lower soil layers remains unclear. Moreover, because it is commonly assumed that nutrient conditions and soil fungal abundance, biomass, and diversity are lower in subsoil than in the topsoil, many researchers have overlooked the relationship between subsoil fungi and grassland degradation (*Wu et al., 2022*). Plants can secrete chemical substances through their roots to alter the soil environment and recruit specific soil microorganisms, thereby creating a unique microbial community in the

rhizosphere (the area surrounding the plant roots). Trampling, grazing, and other behaviors of livestock can alter soil properties, which in turn affect plants communities or that these behaviors directly affect the plant community, changes in these plant communities can lead to modifications in plant root exudates, soil characteristics, and microbial community structures (*Ulbrich et al., 2022*; *Vendruscolo, Mesa & De Souza, 2022*; *Dhungana, Kantar & Nguyen, 2023*). Previous studies have also demonstrated that aboveground plant diversity and community composition are closely linked with soil fungi (*Peng et al., 2020*). As plant roots have different lengths, the influence of plant root exudates on soil fungi varies among different soil depths. Related research has shown that the impact of grazing on soil nutrients in the topsoil and subsoil is not the same (*Zhang et al., 2022*).

We aimed to investigate the effects of different grazing intensities on physicochemical soil characteristics and soil fungi at different depths. We collected soil samples from grasslands that were exposed to different grazing intensities in the Inner Mongolia Autonomous Region, China. By measuring soil physicochemical properties and fungal community composition, we evaluated the interrelationships among grazing intensity, soil properties, and soil fungal communities. We will answer the following two questions: (1) Do fungal communities vary with soil depth and under different grazing intensities? (2) What role do soil physicochemical properties play in regulating the microbial community composition under different grazing intensities? Addressing these questions will help us gain a better understanding of the impact of grazing on soil at different depths.

## MATERIALS & METHODS

### Experimental area
The experimental area is located in Zhurihe Town, Sunite Right Banner, Xilin Gol League, Inner Mongolia, China (112°47′16.9″E, 42°16′26.2″N). The area belongs to the desert steppe ecoregion, with a soil type of light chestnut calcisols. The thickness of the humus layer ranges from 5 to 10 cm. The vegetation in the experimental area is dominated by the species *Stipa breviflora* as foundation species, with *Allium polyrhizum* and *Cleistogenes songorica* as dominant species. Other common species include *Convolvulus ammannii*, *Kochia prostrata*, *Allium tenuissimum*, *Heteropappus altaicus*, *Cleistogenes squarrosa*, *Caragana stenophylla*, and *Carex duriuscula*. The annual precipitation in the experimental area ranges from 170 mm to 190 mm, with an average annual temperature of 5.8 °C, classifying it as a mesothermal climate.

### Experimental design
A total of five treatments were set up in this experiment, with three replicates for each treatment, resulting in 15 experimental grazing plots. Samples were taken at two different locations and two different depths at each test site. A completely randomized block design was employed. The total area of the grazing experimental site was 39.26 hectares. The five grazing treatments were: no grazing (NG), light grazing (LG), moderate grazing (MG), heavy grazing (HG), and over grazing (OG). Based on the local government's guidance documents, as well as the actual grazing practices observed among the herdsmen, the grazing intensities for the various treatment areas were determined. Specifically, the NG

 

treatment had 0 sheep, while the LG, MG, HG, and OG treatments had four, five, six and seven sheep, respectively. These intensities translate into stocking rates of 0 sheep/ha, 1.54 sheep/ha, 1.92 sheep/ha, 2.31 sheep/ha, and 2.69 sheep/ha, respectively.

## Sample collection and index determination

The experimental site was established in 2022, and the grazing period lasts for six months, starting in mid-May and ending in mid-October each year. In May 2023, three different sampling points were randomly selected in each experimental plot. We used a soil auger (with a diameter of 10 cm) to collect soil samples from both the topsoil (0–20 cm) and the subsoil (20–40 cm). Based on previous studies, we discovered that the soil texture undergoes significant changes at a depth of 20 centimeters. Therefore, we selected this depth as the boundary for our study. The three soil samples from the same plot were mixed together to form a single sample, and then passed through a 2-mm sieve to remove plant roots and gravel. The mixed samples were divided into two portions. One portion was placed in sterile centrifuge tubes inside an ice box and transported back to the laboratory for storage at $-20\,°C$ for the determination of soil microbial community structural characteristics. The other portion was placed in a self-sealing bag for the measurement of soil physicochemical properties.

## Index determination of soil physical and chemical properties

To assess soil physicochemical properties, water and soil samples were mixed at a ratio of 2.5:1, and the supernatant of the mixed solution was used to measure the pH using a pH meter. The soil organic matter (SOM) was calculated based on the content of soil organic carbon (SOC), which was determined by oxidation with potassium dichromate and external heating (*Nelson, Sommers & Sparks, 1996*). To assess dissolved organic carbon (DOC) content, water and soil samples were mixed at a ratio of 5:1, filtered through a $0.045\,\mu m$ membrane, and measured using a TOC-L instrument. After extracting the soil sample with potassium chloride, soil nitrate nitrogen ($NO_3^-$-N) and ammonium nitrogen ($NH^{4+}$-N) were determined using a flow analyzer. Soil total nitrogen (TN) was determined using the semi-micro Kjeldahl method. $\beta$-1,4-glucosidase ($\beta$G), N-acetyl-$\beta$-D-glucosidase (NAG), and alkaline phosphatase (AKP) were determined using the microplate fluorescence method (*Marx, Wood & Jarvis, 2001*). SWC was determined by gravimetric method. First measure the weight m0 of the drying aluminum box, put a certain amount of fresh soil into the aluminum box, weigh the weight m1 of the fresh soil and the aluminum box, and measure the weight m2 of the drying soil and the aluminum box after drying.

## DNA extraction and sequencing of soil fungi

The genomic DNA of the sample was extracted using the CTAB method (*Pahlich & Gerlitz, 1980*), and then the purity and concentration of the extracted DNA were detected using agarose gel electrophoresis. Subsequently, an appropriate amount of DNA was taken into a centrifuge tube and diluted to $1ng/\mu l$ with sterile water. Using the diluted genomic DNA as a template, PCR was performed with barcoded specific primers targeting the amplification region and Phusion® High-Fidelity PCR Master Mix (New England Biolabs, Ipswich, MA, USA) to ensure amplification efficiency and accuracy. PCR amplification of the

samples was performed using ITS1 region primers (F: CTTGGTCATTTAGAGGAAGTAA; R: GCTGCGTTCTTCATCGATGC). PCR products were detected using 2% agarose gel electrophoresis. The PCR products were mixed in equal proportions, and then Qiagen Gel Extraction Kit (Qiagen, Hilden, Germany) was used to purify the mixed PCR products. Library preparation was carried out using the NEBNext® Ultra™ II DNA Library Prep Kit. The library quality was evaluated on the Qubit@ 2.0 Fluorometer (Thermo Fisher Scientific, Waltham, MA, USA) and Agilent Bioanalyzer 2100 system. Once the libraries met the required standards, they were sequenced on the NovaSeq 6000 platform. Paired-end reads were merged using FLASH (Version 1.2.11), and the splicing sequences were called Raw Tags. Quality filtering on the raw tags were performed using the fastp (Version 0.20.0) software to obtain high-quality Clean Tags. The Clean Tags were compared with the reference database (Unite database https://unite.ut.ee/ for ITS) using Vsearch (Version 2.15.0) to detect the chimera sequences, and then the chimera sequences were removed to obtain the Effective Tags (*Haas et al., 2011*). The effective sequencing data obtained was denoised using DADA2, and sequences with an abundance less than 5 were filtered out (*Li et al., 2020*), resulting in the final amplicon sequence variants (ASVs). The average number of reads per sample exceeded 35,000. Species annotation was performed using QIIME2 software.

## Data analysis

We used Microsoft Excel 2021 and SPSS to initially organize the data, and used R 4.2.2 to analyze and plot the data. To determine the effects of different grazing intensities and soil depth on soil physicochemical indicators (SOM, TN, ammonium nitrogen, nitrate nitrogen, DOC, pH, and SWC), soil enzyme activities ($\beta$G, AKP, and NAG), and soil fungal communities (abundance and $\alpha$-diversity), a one-way analysis of variance (ANOVA, Tukey test) and T test was conducted. By analyzing these specific soil parameters under different grazing intensities, researchers can gain a better understanding of how these management practices impact soil health and fungal ecology.

Microbial diversity metrics, including total Observed_otus, Chao 1, Shannon diversity, Simpson diversity, Pielou_e, and Goods_coverage, were calculated using the vegan package in R software and visualized using the ggplot2 package. Chao 1 and Observed_otus indices are used to describe species richness, while Shannon and Simpson indices are measures of species diversity. The Pielou_e index is used to describe species evenness. Multivariate PERMANOVAs analysis was conducted using the vegan package in R software. Principal coordinates analysis (PCoA) based on Bray-Curtis distances was performed, and the results were visualized using the ggplot2 package. PCoA could show the differences in fungal communities, and PERMANOVAs analysis proved the statistical characteristics of the differences. It demonstrates the relationship between microbial communities at different depths and under varying grazing intensities. Spearman correlation between soil physicochemical indicators, soil enzyme activity, and soil fungi was calculated using the psych package in R software and visualized accordingly. Heatmaps illustrate the relationship between soil properties and fungal phyla at two soil depths. Multivariate regression trees (MRT) analysis, utilizing the mvpart package in R software, was employed to assess the

most significant abiotic factors influencing fungal $\alpha$-diversity (total observed_otus, Chao 1, Shannon diversity, Simpson diversity, Pielou_e) and community composition (fungal phyla) (*De'Ath, 2002*).

## RESULTS

### Soil characteristics

Grazing affects soil characteristics, but the specific nutrients and the magnitude of differences depend on soil depth. As shown in Table 1, in topsoil, LG had 10.8% higher SOM content than NG, MG had 20.3% lower DOC than NG, LG and HG had higher SWC than NG (35.4% and 50.8% higher, respectively), while LG, MG, and OG had lower AKP content than NG (29.9%, 56.5%, and 30.8% lower, respectively). OG had a 2.8% lower TN content than NG. MG, HG, and OG had lower $\beta$G contents than NG (61.1%, 46.3%, and 61.3% lower, respectively), while all treatments including grazing had lower NAG contents than NG (17.9%, 33.6%, 40.5%, and 74.6% lower, respectively. In the subsoil, HG had higher SOM and TN than NG (12.3% and 10.0% higher, respectively). LG, HG and OG had lower AKP contents in the soil than NG (15.8%, 42.9%, and 45.2% lower, respectively), and the four treatments involving grazing had lower $\beta$G contents than NG (30.0%, 39.7%, 24.9%, and 39.1% lower, respectively, while the NAG contents were 46.1%, 41.3%, 50.7%, and 60.3% lower than NG, respectively. OG had higher nitrate nitrogen that NG (308.2% higher in the topsoil and 466.2% higher in the subsoil). The SWC of the subsoil was higher than that of the topsoil in NG, LG, MG, and HG treatments, but NAG showed the opposite trend. Except for MG, AKP had the same trend as NAG. The content of DOC in surface soil of HG was significantly higher than that of subsurface soil. The $\beta$G content in surface soil of control treatment and grazing treatment was higher than that of subsurface soil. Soil depth reduces the effect of grazing on soil physicochemistry.

### Fungal $\alpha$-diversity

Like for the soil characteristics, the effects of grazing on fungal $\alpha$ diversity were different at different depths. In the topsoil, the effect of grazing on fungal $\alpha$-diversity was not significant (Fig. 1). In the subsoil, the fungal $\alpha$ diversity indices in all treatments involving grazing were lower than those in the plots without grazing, and the Observed_otus index, and Chao1 index in LG, MG and HG were significantly lower than those in the NG treatment (Fig. 1). There were no differences among treatments involving grazing. The Pielou index and Simpson index were also lower, but not significantly (Fig. 1). The Shannon indices of LG and OG were significantly lower than the index for NG (Fig. 1). For both topsoil and subsoil, all grazing treatments reduced $\alpha$ diversity indices of the subsoil fungi. Grazing also changed the effect of depth on fungal $\alpha$ diversity. In the NG treatment, the Observed_otus index and Chao1 index in the topsoil were lower than those in the subsoil (Fig. 1). However, there were no significant differences in fungal $\alpha$ diversity at different depths under the different treatments with grazing.

### Fungal $\beta$-diversity

Grazing had a large effect on the relative abundance of fungi, and the effects of grazing on fungal relative abundance were different at different depths. The PCoA sorting results

**Table 1  Effects of different grazing intensities on soil and microbial characteristics at two soil depths.**

| Depth | 0–20 cm | | | | | | |
|---|---|---|---|---|---|---|---|
| Treatment | NG | LG | MG | HG | OG | F | P value |
| OM (g/kg) | 19.09[bA] | 21.16[aA] | 18.94[b] | 18.60[bB] | 17.98[b] | 10.32 | <0.001 |
| TN (g/kg) | 0.98[aA] | 1.04[aA] | 0.98[aA] | 0.99[a] | 0.90[b] | 5.09 | 0.004 |
| $NH_4^+$ (mg/kg) | 2.61 | 2.48[A] | 2.67 | 2.60 | 2.72 | 0.82 | 0.528 |
| $NO_3^-$ N(mg/kg) | 1.34[b] | 1.14[bA] | 1.50[b] | 2.09[b] | 5.47[a] | 3.00 | 0.038 |
| DOC (mg/kg) | 64.77[ab] | 58.19[bc] | 51.63[c] | 72.23[aA] | 64.23[ab] | 6.40 | 0.001 |
| $\beta$G (nmol/g/h) | 200.65[aA] | 170.05[aA] | 77.98[bc] | 107.67[b] | 77.71[c] | 27.00 | <0.001 |
| NAG (nmol/g/h) | 57.87[aA] | 47.53[bA] | 38.41[cA] | 34.45[cA] | 14.69[d] | 31.73 | <0.001 |
| AKP (nmol/g/h) | 199.71[aA] | 140.09[bA] | 86.96[cA] | 175.99[aA] | 138.19[b] | 17.49 | <0.001 |
| pH | 8.18 | 8.25 | 8.26 | 8.28[A] | 8.30[A] | 0.67 | 0.618 |
| SWC (%) | 1.30[cA] | 1.76[abA] | 1.96[aA] | 1.49[bcA] | 1.41[c] | 6.07 | 0.001 |

| Depth | 20–40 cm | | | | | | |
|---|---|---|---|---|---|---|---|
| Treatment | NG | LG | MG | HG | OG | F | P value |
| OM (g/kg) | 17.84[bB] | 18.11[bB] | 18.37[b] | 20.03[aA] | 18.53[b] | 4.17 | 0.01 |
| TN (g/kg) | 0.90[bB] | 0.89[bB] | 0.89[bB] | 0.99[a] | 0.95[ab] | 3.81 | 0.015 |
| $NH_4^+$ (mg/kg) | 2.64 | 2.84[B] | 2.84 | 2.68 | 2.84 | 1.09 | 0.384 |
| $NO_3^-$ N (mg/kg) | 1.39[b] | 1.44[bB] | 2.17[b] | 3.86[b] | 7.87[a] | 4.22 | 0.01 |
| DOC (mg/kg) | 58.73 | 56.00 | 53.90 | 58.60B | 57.69 | 0.61 | 0.66 |
| $\beta$G (nmol/g/h) | 122.25[aB] | 85.62[bB] | 73.72[b] | 91.82[b] | 74.44[b] | 8.047 | <0.001 |
| NAG (nmol/g/h) | 33.50[aB] | 18.06[bB] | 19.65[bB] | 16.53[bB] | 13.30[b] | 10.14 | <0.001 |
| AKP (nmol/g/h) | 121.49[aB] | 102.33[bB] | 121.49[aB] | 69.42[cB] | 66.60[c] | 27.93 | <0.001 |
| pH | 8.23 | 8.26 | 8.27 | 8.36[B] | 8.24[B] | 1.92 | 0.138 |
| SWC (%) | 2.19[B] | 2.14[B] | 2.44[B] | 2.34[B] | 1.95 | 1.87 | 0.148 |

**Notes.**

For each parameter, a different letter indicates a significant difference at the 0.05 probability level ($P < 0.05$) based on Tukey's HSD. F and P values in bold show statistically significant differences.

Uppercase letters indicate differences between different depths, and lowercase letters indicate differences between different grazing intensities. No letter indicates no significant difference.

NG, no grazing; LG, light grazing; MG, moderate grazing; HG, heavy grazing; OG, over grazing; OM, organic matter; TN, soil total nitrogen content; DOC, dissolved organic carbon; $\beta$G, $\beta$-1,4-glucosidase; NAG, $\beta$-1,4-Nacetylglucosaminidase; AKP, alkaline phosphatase; SWC, soil water content.

showed that there were significant differences in fungal community composition (beta diversity) in the treatments involving grazing than in the NG. There were also significant differences in fungal community composition between light, moderate, and heavy grazing (Fig. 2). There were no differences in fungal community composition between the two soil depths, similarly, bifactor analysis in grazing and soil depth did not find significant differences (Fig. 2).

The dominant phyla were *Ascomycota*, *Basidiomycota*, *Mortierellomycota* and *Glomeromycota*. In the topsoil, the relative abundance of dominant fungi was not significantly different. In the subsoil, the relative abundance of *Mortierellomycota* was lower in LG, HG, and OG than in the NG by 82.7%, 61.9%, and 47.7%, lower, respectively; (Fig. 2B). *Glomeromycota* was 51.5% higher in HG than NG (Fig. 2B). Compared with topsoil, *Glomeromycota* in subsoil was reduced by 20.4% (Table S3). However, the relative

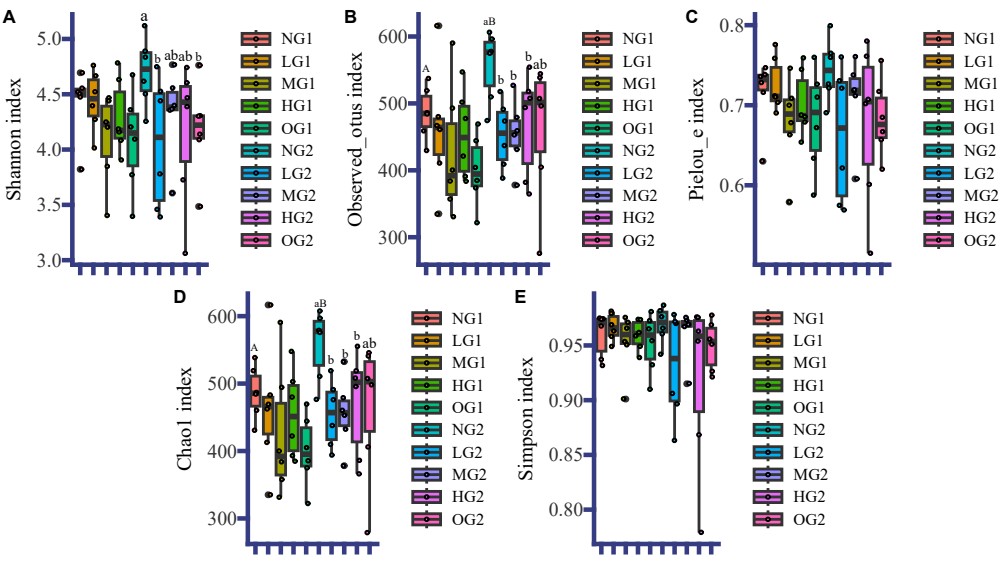

**Figure 1** **Fungal α diversity.** NG1, no grazing for topsoil; LG1, light grazing for topsoil; MG1, moderate grazing for topsoil; HG1, heavy grazing for topsoil; OG1, over grazing for topsoil; NG2, no grazing for subsoil; LG2, light grazing for subsoil; MG2, moderate grazing for subsoil; HG2, heavy grazing for subsoil; OG2: over grazing for subsoil. We used the $t$-test to analyze the data. Uppercase letters indicate differences between different sampling depths, and lowercase letters indicate differences between different grazing intensities. No letter indicates no significant difference. Values are means. The statistics are provided in Table S5.

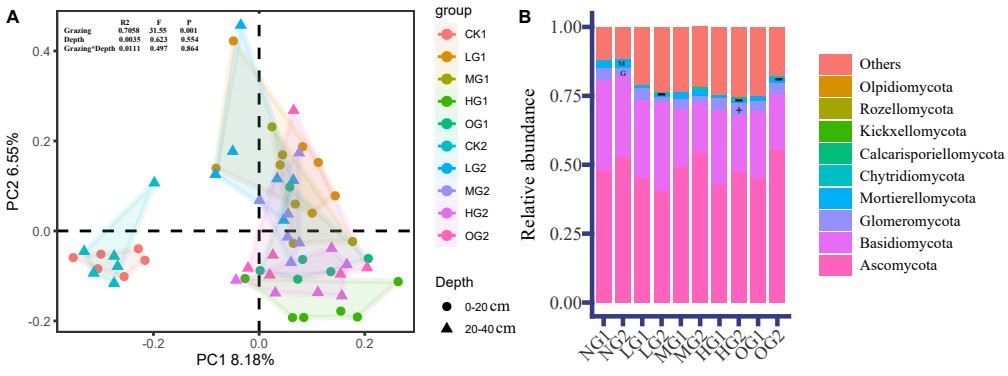

**Figure 2** **Fungal β diversity (PCoA) (A) and relative abundance of fungi (B).** The PCoA construction was performed by using Amplicon Sequence Variants. G, Glomeromycota; M, Mortierellomycota. The symbols '+' and '−' indicate a significant ($P < 0.05$) increase and decrease, respectively (Table S4). NG1, no grazing for topsoil; LG1, light grazing for topsoil; MG1, moderate grazing for topsoil; HG1, heavy grazing for topsoil; OG1, over grazing for topsoil; NG2, no grazing for subsoil; LG2, light grazing for subsoil; MG2, moderate grazing for subsoil; HG2, heavy grazing for subsoil; OG2, over grazing for subsoil.

abundance of *Chytridiomycota* and *Calcarisporiellomycota* in subsoil was 218% and 337%, respectively, of that in topsoil (Table S3).

At the genus level, in the topsoil, compared with NG, HG and OG make the relative abundance of *Coprinopsis* decreased by 99.5% and 98.8%, respectively (Table S1). LG,MG,HG and OG significantly decreased the relative abundance of *Clitocella* (Table S1). LG and MG significantly decreased the relative abundance of *Limonomyces* (Table S1). HG significantly increased the relative abundance of *Ceratobasidium* (Table S1). OG caused a significant decline in the relative abundance of *Cercophora* (Table S1).

In the subsoil, compared with NG, LG and HG significantly decreased the relative abundance of *Mortierella* and *Cercophora* (Table S1). HG significantly increased the relative abundance of *Glomus* and *Ceratobasidium* (Table S1). LG,MG,HG,OG significantly decreased the relative abundance of *Preussia* and *Clitocella* (Table S1). OG significantly decreased the relative abundance of *Coprinopsis* (Table S1). LG significantly decreased the relative abundance of *Darksidea* and *Limonomyces* (Table S1). However, there was no significant difference in the relative abundance of fungi genera at different soil depths.

### The linkage between soil characteristics and fungal communities

The correlation between fungi and soil environmental factors varied with soil depth, as heatmap analysis revealed significant relationships between soil physicochemical properties and fungi, with distinct patterns observed in the topsoil and subsoil. The topsoil and subsoil showed different patterns. In the topsoil, *Chytridiomycota* were significantly correlated with DOC, $\beta$G, AKP, and SWC, *Basidiomycota* were significantly positively correlated with AKP, while *Kickxellomycota* were significantly positively correlated with pH (Fig. 3). *Calcarisporiellomycota* were significantly positively correlated with SWC (Fig. 3). In the subsoil, pH significantly correlated with *Basidiomycota*, *Glomeromycota*, and *Calcarisporiellomycota*, while AKP had a significant negative correlation on *Glomeromycota* (Fig. 3). *Rozellomycota* had a significant negative correlation with TN and SOM (Fig. 3). *Basidiomycota* had a significant positive correlation with NAG (Fig. 3). *Calcarisporiellomycota* were significantly positively correlated with SWC (Fig. 3).

Through multiple regression tree analysis, the main driving factor regulating fungal community changes appeared to be SWC in the topsoil and ammonium nitrogen and nitrate nitrogen in the subsoil (Figs. 4A and 4B). For changes in fungal $\alpha$-diversity, TN was the main factor influencing the topsoil, while nitrate nitrogen had a significant impact on the subsoil (Figs. 4C and 4D).

## DISCUSSION

### Impact of grazing on soil physiochemical properties

Our results demonstrate that the impact of different grazing intensities on topsoil and subsoil is not entirely similar. In the topsoil, soil organic matter (SOM) content was affected by grazing, with first an increasing and then a decreasing trend with an increase in intensity, reaching its maximum under light grazing. This finding is consistent with previous research (*Song et al., 2023*), which indicated that moderate grazing can enhance SOM content. Light grazing can have a positive effect on soil organic matter content through production of manure, increase decomposition in the plant litter, and activation of nutrients by root exudates (*Inagaki et al., 2023*; *Mikutta et al., 2019*; *Sokol, Sanderman*

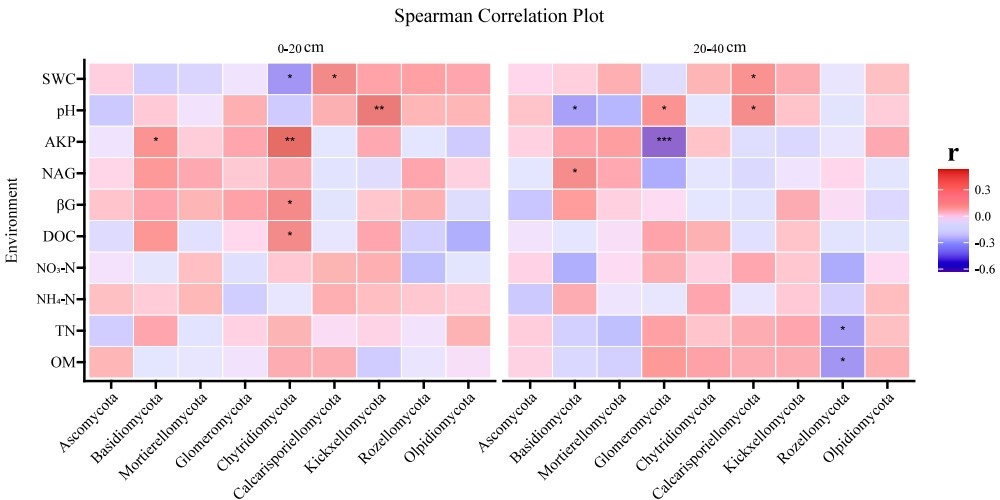

**Figure 3** **The correlation analysis of environmental factors and fungi (at phylum level) at two soil depths.** Spearman's correlation analysis between soil physicochemical properties and fungal $\alpha$ diversity at two soil depths; red represents positive correlation and blue represents negative correlation. *Significant at the 0.05 probability level. **Significant at the 0.01 probability level. ***Significant at the 0.001 probability level.

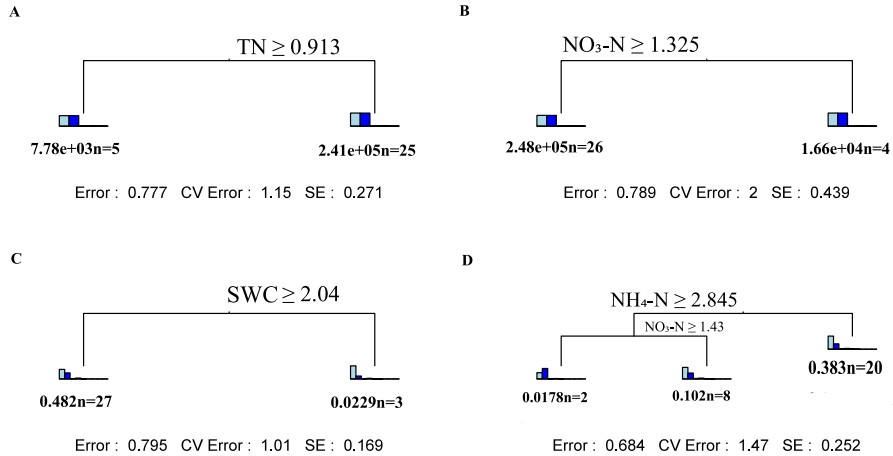

**Figure 4** **Multivariate regression tree analysis of the impact of environmental factors on soil fungal community composition and $\alpha$ diversity.** Multivariate regression tree analysis: (A) Shallow layers soil fungi $\alpha$ diversity; (B) deep layers soil fungi $\alpha$ diversity; (C) shallow layers soil fungi $\alpha$ community; (D) deep layers soil fungi $\alpha$ community. Inside the bar graph in A and B from left to right: Chao1, Observed_otus, Pielou_e, Shannon, Simpson. Inside the bar graph in C and D from left to right: *Ascomycota*, *Basidiomycota*, *Mortierellomycota*, *Glomeromycota*, *Chytridiomycota*, *Calcarisporiellomycota*, *Kickxellomycota*, *Rozellomycota*, *Olpidiomycota*.

*& Bradford, 2019*). Our results show grazing reduce the plant litter (Table S2). Grazing livestock can impact plants through various behaviors such as feeding and rubbing, which causes changes in plant litter. Additionally, trampling by livestock accelerates the

decomposition of litter in the soil. However, as grazing intensity increases, its effect on litter decomposition gradually decreases (*Su et al., 2022*; *Wei et al., 2023*).

Previous studies revealed that the impact of grazing intensity on soil total nitrogen content is nonlinear (*Dai et al., 2022*). In grasslands, the total nitrogen content initially increases with grazing intensity and then decreases, reaching a maximum under light grazing (*Shi et al., 2023*; *Cao et al., 2024*). This is not completely consistent with our results, which may be related to the type of grassland or the climate environment, but the overall trend is consistent. Livestock manure and plant litter are two main sources of soil nitrogen (*Wang et al., 2020*), the decomposition rate of manure and litter in the soil decreases as grazing intensity increases. Nitrate nitrogen in the soil mainly originates from the urine of livestock. Previous studies showed that as grazing intensity increases, the content of soil nitrate continuously increases (*Li et al., 2023a*; *Li et al., 2023b*; *Zhang et al., 2023a*; *Zhang et al., 2023b*; *Lv et al., 2024*). This is consistent with our results. An increase in grazing intensity leads to a decrease in plant biomass (Table S2), leading to decreased absorption of nitrate by plants (*Kant, 2018*), Additionally, grazing can also reduce the loss of nitrate from the soil (*Eriksen et al., 2015*), resulting in accumulation of nitrate in the soil.

Compared with NG, soil DOC content became lower under LG, MG, and OG conditions, but higher under HG. Soil DOC originates from plant litter, and the effects of grazing on DOC are similar to the above effects of grazing on SOM. The influence of soil moisture is more pronounced during periods of low precipitation (*Luo et al., 2009*). Our results show Grazing increased soil water content, reaching a maximum under MG. Grazing increases surface litter, but livestock trampling accelerates the decomposition rate of the litter. Previous studies found that the increase in litter can reduce soil water evaporation, thereby increasing SWC (*Zhao et al., 2023*). Due to the effect of litter and water evaporation, soil water content reaches its maximum under MG.

In the subsoil, the trends of nitrate, DOC, SWC, SOM, and TN were consistent with those in the topsoil. However, the content of SOM and TN reached its maximum under HG, rather than under LG. This is because the SOM in surface soil layers mainly originates from root exudates and not from plant litter (*Jobbágy & Jackson, 2000*), and the impact of livestock on soil decomposition capacity, jointly determine this result. Plant growth requires nitrogen (N) absorption through their roots, and moderate grazing promotes plant growth (*Song et al., 2022*). Therefore, under LG and MG, TN content decreases due to the utilization of N by plants. But when the number of plants falls to a certain number, the reproduction ability declines. Therefore, under HG and OG, excessive herbivory by livestock leads to a decrease in the number of plants, reducing the absorption of N and resulting in an increase in TN content (Table S2). Compared to the topsoil, the subsoil generally has lower nutrient content, with the exception of SWC. This is due to our experimental site being located in a desert steppe, where the water-holding capacity of subsoil is stronger than that of the topsoil. For nitrate, DOC, and SWC, the influence of grazing is mitigated by soil depth, as livestock activities have a greater impact on topsoil than on subsoil.

## Impact of grazing and soil depth on fungi

Fungi are heterotrophic organisms that primarily feed on decaying substrates. Grazing has an impact on both the quality and quantity of litter, which can easily alter fungal diversity in grasslands (*Rong et al., 2022*). Grazing treatments can also affect fungi by altering the interactions between soil fungi and their habitats (*Guitian & Bardgett, 2000*; *Schipper et al., 2014*; *Zhang et al., 2020*). Our results show that various $\alpha$-diversity indices of soil fungi under treatments with grazing are lower than those under the NG treatment, which is inconsistent with previous findings (*Wang & Tang, 2019*). On the one hand, grazing alters the soil environment, leading to changes in fungal diversity. For example, grazing altered SOC thus affecting soil microbial communities (*Zhang et al., 2024*). On the other hand, grazing treatment modifies plant species composition, resulting in changes in the fungal community recruited through root exudates (*Xu et al., 2022*). Changes in plant species composition, such as tomato and onion intercropping, increased the colonization of specific *Bacillus spp* and established specific rhizosphere microbial communities compared with tomato cultivation alone (*Zhou et al., 2023*). The various $\alpha$-diversity indices in subsoil were higher than those in topsoil. These results are consistent with previous reports (*Wang et al., 2023*). This is because, compared to topsoil, fungi in subsoil are stronger influenced by root exudates. During plant growth, root exudates exert a recruitment effect (*Zhou et al., 2023*), resulting in a higher $\alpha$-diversity of fungi in subsoil than in topsoil.

In our study, the primary dominant fungal phyla in all soils were *Ascomycota*, followed by *Basidiomycota*, *Glomeromycota*, and others. Our experimental site belongs to a desert steppe, where fungi require stronger decomposition abilities to obtain organic matter than in other environments. *Ascomycota* are the primary decomposer among soil fungi (*Inagaki et al., 2023*). *Ascomycota* is the most dominant fungal phylum, with a relative abundance exceeding 40% in all treatments. Furthermore, *Ascomycota* are more abundant in subsoil than in topsoil. The reason for this is that topsoil is rich in humus, which is easy to decompose. Compared to topsoil, subsoil contains a greater variety of recalcitrant substances. Therefore, more *Ascomycota* are required in subsurface soil to decompose these substances (*Manici et al., 2024*). Since subsoil has less soil OM than topsoil, oligotrophic fungi such as *Basidiomycota* are more abundant in topsoil, while copiotrophic fungi such as *Ascomycota* are more abundant in subsoil (*Yang et al., 2023*). Grazing alters the relative abundance of fungi. For instance, under LG, HG, and OG treatments, the relative abundance of *Mortierellomycota* was significantly lower due to changes in the soil environment caused by grazing (*Ahonen et al., 2024*). Additionally, with increasing grazing intensity, certain fungal genera belonging to the same phylum exhibited opposite changes. For example, within *Basidiomycota*, *Ceratobasidium* and *Coprinopsis* displayed opposite trends. This may be due to the fact that these two fungal genera within the *Basidiomycota* phylum play similar ecological roles in the soil (*Snajdr et al., 2011*), but they prefer different soil environments for growth.

### Regulating effects of soil properties on the response of fungi to grazing intensity

Multivariate regression tree (MRT) analysis revealed that soil total nitrogen (TN) was the primary factor driving changes in fungal $\alpha$-diversity in the topsoil, while nitrate nitrogen was the main factor in the subsoil. Previous studies also demonstrated that nitrogen elements in the soil significantly influence changes in soil fungi (*Chen et al., 2021*). In the subsoil, $\beta$G showed a positive correlation with the Chao1 and Observed_otus indices (Table S1). $\beta$G is a key enzyme for microbial carbon acquisition (*Sinsabaugh et al., 2008*). Compared with topsoil, fungi in the subsoil require more carbon to maintain their biological functions (*Huang et al., 2023*). The MRT analysis revealed that the main driving factor regulating fungal community changes was SWC in the topsoil, while it was ammonium nitrogen and nitrate nitrogen in the subsoil. For the topsoil, previous studies have shown that soil microorganisms are very sensitive to nutrients and soil water, and even small changes in water content can lead to changes in the dominant members of the soil microbial community (*Ramírez et al., 2020*; *Cao et al., 2022*). When soil water content decreases, soil aeration increases (*Levy-Booth et al., 2016*); this may affect the relative abundance of anaerobic and aerobic bacteria in the soil. In the subsoil, nutrients such as nitrogen have a strong influence on the fungal community due to the lack of nutrients.

In our correlation analysis, Glomeromycota in subsoil showed a strong negative correlation with AKP (alkaline phosphatase) content and a positive correlation with pH. Glomeromycota, also known as arbuscular mycorrhizas fungi, can promote plant absorption of phosphorus (*Bushra et al., 2024*). The decrease in phosphorus (P) content affects the AKP levels in the soil (*Gao et al., 2024*). Additionally, plants absorb P from the soil in the form of phosphates and secrete various substances to enhance soil nutrient content (*Oldroyd & Leyser, 2020*), which in turn promotes fungal growth. With increasing soil depth, the correlation of soil characteristics such as DOC, SWC, AKP and $\beta$G with *Chytridiomycota* disappeared, and negative correlations of TN and SOM with *Rozellomycota* emerged. This may be because changes in the nutrient environment result in altered fungal life strategies (*Yang et al., 2023*). The change in fungal life strategy liberates the fungus from being previously limited by nutrients or creates new constraints. Nutrient-rich fungi such as *Chytridiomycota* typically exhibit a positive correlation with soil nutrients when soil nutrient content is low. However, in conditions of abundant soil nutrients, they may be subject to self-limitation. The oligotrophic fungi such as Rozellomycota, on the other hand, show a negative correlation with soil nutrients when soil nutrient content is high.

## CONCLUSIONS

This study showed that soil properties and microbial changes in the topsoil and subsoil responded differently to different grazing intensities. In the topsoil, SOM and TN was higher under LG than in the no grazing treatment, while DOC content was higher under heavy grazing. MG, HG, and OG showed $\beta$G content than NG, while LG, MG, and OG had lower AKP contents than NG. The NAG content was lower under all four grazing intensities than under in NG control. In the subsoil, SOM and TN were higher under

HG. LG, HG, and OG had lower AKP contents than NG. All four grazing intensities had lower $\beta$G and NAG contents than the no grazing treatment. Grazing reduces the fungal $\alpha$-diversity indices (Shannon, Chao1, and OTUs) in subsoil. Grazing intensity resulted in changes in soil fungal community structure. In the subsoil, light, heavy, and excessive grazing resulted in significantly lower relative abundance of *Ortierellomycota*, while heavy grazing greatly enhanced the relative abundance of *Glomeromycota*. With increasing soil depth, the relative abundance of *Ascomycota*, the dominant fungal phylum, increased. However, this trend was reversed under LG. Changes in soil depth altered the relationship between soil properties and fungi. The results of our study indicate that grazing treatments regulate fungal community changes in topsoil by altering SWC, while in subsoil, they regulate fungal community composition by modifying ammonium nitrogen and nitrate nitrogen levels. Specific soil fungal communities were produced under different grazing intensities, and these specific fungal communities had different effects on soil sustainable use. Grazing changes soil fertility through direct or indirect effects on soil nutrients. Our research provides empirical evidence of the impact of grazing on soil fertility in desert steppes.

## ACKNOWLEDGEMENTS

Thanks to Pengyu Qu and Xiaojiang Yang for their help in sample collection.

### Funding

The study was financially supported by the National Natural Science Foundation of China (32071861, 42077054). The study was also supported by the National Key R&D Program of China (2022YFD1900300), and the National Natural Science Foundation of Inner Mongolia (2022LHMS03004, 2022MS03071). The funders had no role in study design, data collection and analysis, decision to publish, or preparation of the manuscript.

### Grant Disclosures

The following grant information was disclosed by the authors:
The National Natural Science Foundation of China: 32071861, 42077054.
The National Key R&D Program of China: 2022YFD1900300.
The National Natural Science Foundation of Inner Mongolia: 2022LHMS03004, 2022MS03071.

### Competing Interests

The authors declare there are no competing interests.

### Author Contributions

- Xiangjian Tu conceived and designed the experiments, performed the experiments, analyzed the data, prepared figures and/or tables, authored or reviewed drafts of the article, and approved the final draft.

- Paul C. Struik analyzed the data, authored or reviewed drafts of the article, and approved the final draft.
- Shixian Sun performed the experiments, prepared figures and/or tables, authored or reviewed drafts of the article, and approved the final draft.
- Zhang Wenbo performed the experiments, analyzed the data, prepared figures and/or tables, and approved the final draft.
- Yong Zhang performed the experiments, analyzed the data, authored or reviewed drafts of the article, and approved the final draft.
- Ke Jin conceived and designed the experiments, authored or reviewed drafts of the article, and approved the final draft.
- Zhen Wang conceived and designed the experiments, authored or reviewed drafts of the article, and approved the final draft.

## DNA Deposition

The following information was supplied regarding the deposition of DNA sequences:

The sequences are available at GenBank: NG_070110.1 and NR_171887.1.

## Data Availability

The raw measurements are available in the Supplementary Files.

## Supplemental Information

Supplemental information for this article can be found online at http://dx.doi.org/10.7717/peerj.18791#supplemental-information.

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
