# Peer review of "Responses of fungal communities at different soil depths to grazing intensity in a desert steppe"

_PeerJ, doi:10.7717/peerj.18791_

## Round 0.1 · original submission · Major Revisions

The reviewers suggested different approaches to statistical analysis. These suggestions should be carefully considered. Additionally, I have further concerns regarding some of the current statistical analyses, as outlined below. The reviewers have requested various other revisions and corrections, and I have provided my own specific comments as follows:

Abstract – I didn’t understand the last sentence. What do you mean by “addressing”?
L78-79 “The impact…is not linear” – can you cite a reference to support this statement?
L 119 -120 In place of “Are there specific fungal communities at different…” , consider “Do fungal communities vary with soil depth and under different grazing intensities?”
L 130 and elsewhere – Species names need to be in italics
L 145-147 I agree with reviewers that these seem like small, incremental differences in grazer density. Please explain how these were assigned names like “heavy”, “overgrazed” , etc.
L 198 – Duncan’s test is generally not accepted as a statistical test in ecology or many other fields because it does not provide expected control of Type 1 error rates during multiple comparisons. This was clearly demonstrated in simulations by Day and Quinn (1989):
https://www.jstor.org/stable/1943075
Free PDF link: https://sciences.ucf.edu/biology/pascencio/wp-content/uploads/sites/24/2016/11/comparisons_of_treatments_in_ANOVA.pdf
More recently, this has been re-enforced by Midway et al 2020:
https://peerj.com/articles/10387/
Duncan’s test is specifically “not recommended” (see Table 2). Tukey HSD is commonly available in most statistics packages and is commonly recommended for controlling experiment-wide Type 1 error, although other recommended multiple comparison methods listed in Table 2, like sequential Bonferroni, may also be appropriate. Note that if you convert grazing to continuous variable as suggested by reviewer 3 and analyze the shape of the response then multiple comparisons among treatment categories may no longer be relevant.

ANOVAs for soil properties and fungi – these tests require normality and homogeneity of variance for residuals across treatment. Was this tested? I am guessing homogeneity of variance has been strongly violated based on relatively low variance in the NG treatment and much higher variances in the grazing treatments seen in the graphs in Figure 1. P-values for ANOVA and multiple comparisons may not be trustworthy in that case and transformation or other approaches may be needed if you want to make inferences based on P-values.

Figure 2B – What is the ordering of taxa in the legend? At first I thought it was from most to least abundant in NG, but that is not the case.
Figure 2 caption – please indicate what variables are included in the PCA and which factors are most important along each of the PCA axes. I thought the relative abundances of taxa from Fig 2B were used to make the PCA, but that PCA shows clear differentiation between NG and other treatments, while the relative abundance of taxa in Fig 2B don’t differ obviously between NG and all other treatment, except in the category “other”. Is PCA axis 2 strongly determined by “other” taxa, explaining the separation seen in Fig 2A between NG and the other treatments?
Figure 3 caption – please indicate the meaning of the symbols found inside some of the squares.
Figure 4 caption – please indicate the meaning of the shaded color blocks and meaning of the numbers immediately beneath the colors.

L 302 What do you mean by “rubbing”?
Discussion – you highlight changes in various specific taxa below the Phyla level in the Results section, but none of them are mentioned in the Discussion. Do any of the specific below-Phyla taxa changes merit Discussion? If none of them are interesting or important, then I wonder why they are highlighted in the Results?
L 372 Which two fungal genera?
L 428 You might consider specifying the type of help provided (?)


Reviewer 1 ·

Basic reporting

Introduction to the Impact of Grazing on Soil Microbes: The introduction does a good job explaining how grazing alters soil environments, thereby affecting soil microbes. However, it is also important to establish that grazing-induced changes in plant diversity and composition can also affect soil microbial communities. I suggest revising the introduction to at least mention this important point, which is later discussed in the manuscript.

Consistency in Terminology and Organization: Given the wealth of data collected and the metrics calculated, I urge the authors to revise the manuscript for consistency in terms used and the presentation of results. For example, in the results and discussion sections, the terms SOM and OM are used interchangeably; I suggest using one term consistently unless a crucial distinction needs to be made. Similarly, terms like diversity, alpha diversity, richness, and their associated metrics are used interchangeably; please revise for clarity.

Additionally, when presenting results, the authors should be consistent in the order in which they discuss findings (e.g., topsoil first, then subsoil) and ensure parallel comparisons throughout. I suggest reorganizing the results section to present the most important (big picture) results directly linked to the research question (e.g., grazing affects the diversity and composition of soil fungal communities, but the magnitude and direction of changes, as well as the specific taxa affected, differ between topsoil and subsoil) and then delve into the details for each soil depth. I also urge the authors to ensure that figures are easy to interpret and directly associated with the statistical tests conducted (e.g., Figure 2B: stacked bar graphs that do not show variation among samples may not accurately reflect the results of Duncan's test).

Comparative Analysis of Fungal Communities: related to the comment above, one of the study's questions was whether there are specific fungal communities at different soil depths under varying grazing intensities. However, the first part of this question, "differences among soil depths," seems to be lost in the results presentation, as results are mainly discussed separately for topsoil and subsoil. I suggest that in their revision, the authors organize the results so that it is clear when they are comparing topsoil and subsoil and when comparing grazing treatments within each soil depth.

Experimental design

Evaluation of Soil Characteristics and Fungal Communities: When analyzing the association between soil characteristics and fungal communities (Lines 273-284), the authors evaluated subsoil and topsoil separately and found that the patterns of association differed between the two depths, even within the same phylum. This approach is confusing for several reasons:
o Soil characteristics evaluated (e.g., SOM, DOC, SWC, pH) are continuous variables, and thus their association with fungal communities should be evaluated as a whole, regardless of soil depth, especially since the authors have already established differences in soil characteristics between depths and grazing treatments.
o At Lines 399-403, the authors suggest that the correlation between certain fungal phyla and specific soil characteristics changes with depth. While this may be true, several factors could explain this, including differences in the specific taxa within each phylum between topsoil and subsoil, or the possibility that certain soil nutrients do not vary at one depth but do at the other.
Please revise to clarify this, either by explaining the rationale behind your approach, reanalyzing the data using each nutrient as a continuous variable regardless of soil depth, or analyzing lower taxonomic levels within each phylum.


Statistical Analysis Reporting: Throughout the manuscript, the authors do not adequately report the results of the main statistical tests performed (e.g., p-values and F statistics), often only providing results of post-hoc pairwise comparison tests (i.e., Duncan's test; Table 1 and Figure 1). I urge the authors to provide more details on the statistical tests performed, describing the model evaluated, the fixed and random effects considered, and reporting the main results of these tests (possibly as supplementary tables or in the text), which would aid in understanding the results in the context of the research question.

Validity of the findings

Evaluation of Interrelationships Among Factors: One of the key premises of the study, as stated in the introduction, was to evaluate "the interrelationships among grazing intensity, soil properties, and soil fungal communities." While this point is touched upon in the discussion, the analyses and results do not directly address this question, instead treating each factor separately. This approach weakens the discussion on the interrelationships among these factors, despite the data being available to do so.
o Alpha Diversity Metrics: I suggest employing a path analysis, structural equation modeling (SEM), or a similar approach to evaluate the direct (grazing → fungal diversity) and indirect (grazing → soil characteristics → fungal diversity) effects of grazing on fungal diversity. To reduce complexity, the authors could focus on soil characteristics identified as primary drivers of composition and diversity in their MRT (Figure 4) or Spearman’s correlation (Figure S1). Relevant literature includes Henning et al 2021 Ecology (https://esajournals.onlinelibrary.wiley.com/doi/full/10.1002/ecy.3210).
o Beta Diversity: I recommend rerunning the PERMANOVA or a similar test to include soil characteristics in the model and evaluate the interaction terms between grazing and soil characteristics. Complexity could be reduced by focusing only on the nutrients identified as primary drivers of fungal diversity and composition as above.

Microbial Composition Analysis at the Phylum Level: The discussion on microbial composition results primarily focuses on the phylum level using the copiotroph–oligotroph framework. While differences at the phylum level are interesting, it is puzzling why this level was chosen. Studies such as For example, Ho et al. 2017 FEMS (https://academic.oup.com/femsec/article/93/3/fix006/2937747) and Sauvadet et al. 2019 Soil Biology and Biochemistry (https://www.sciencedirect.com/science/article/pii/S0038071719300306) have shown significant variation in physiological traits within phyla, suggesting limitations in using this model at the phylum level. Further, Ndinga-Muniania et al. 2023 PloS One (https://journals.plos.org/plosone/article?id=10.1371/journal.pone.0287990) have demonstrated that growth and resource use traits in fungi are mainly conserved at the order or, to some extent, class level. I suggest either explaining the rationale for this approach or revising the manuscript to include analyses and discussions at lower taxonomic levels using the same framework.

Clarification of Nutrient Levels and Fungal Abundance: The authors suggest that subsoils have fewer nutrients than topsoils, which is why Ascomycota (copiotrophic) are more abundant in subsoils and Basidiomycota (oligotrophic) are more abundant in topsoils (Lines 364-366). This is confusing because copiotrophic taxa are typically defined as fast-growing microorganisms that thrive in soils with high carbon mineralization rates, whereas oligotrophic taxa are slow-growing and predominate in soils with more recalcitrant carbon substrates ( see, Fierer et al, 2007 Ecology, https://esajournals.onlinelibrary.wiley.com/doi/full/10.1890/05-1839). Please revise and clarify this, considering your results of soil nutrient.

Additional comments

Line 67: Replace "theoretical" with "empirical" to better reflect the nature of the study.

Line 99-101: Rephrase to clearly convey that studies of microbial diversity and function have primarily focused on topsoil.

Line 150-154: Clarify the number of samples compared in this study. As currently written, it sounds like a total of 30 samples (5 treatments x 3 replicates x 2 soil depths) were examined. If so, please revise here and elsewhere in the manuscript for clarity.

Line 188-191: The description of sequence analyses needs more detail. Specifically, I suggest revising this section to include details on taxonomic assignment of sequences, whether non-fungal reads were assigned, if so, how many, and whether they were discarded, as well as the average number of reads per sample or the range.

Line 203-207: The authors do a good job explaining what each of the calculated indices measures, except for "Goods_average." Given that this metric is not referred to frequently in the manuscript, I suggest either deleting it to reduce complexity or indicating what it measures.

Line 221-237: As currently written, this section, which highlights the impact of grazing on soil characteristics, may lead to confusion and be difficult to understand. Mainly, the authors only report on treatments that were significantly different from the NG, neglecting to report differences (or lack thereof) among grazing treatments, and the text jumps to specific nutrients without providing a "big picture" result.

I suggest revising this section by organizing the ideas in terms of the most important (big picture) results directly linked to the research question (i.e., grazing affects soil characteristics, but the specific nutrients and the magnitude of differences depend on soil depth), along with the appropriate statistical tests. Then, discuss the most important soil characteristics for each depth, describing results in relation to grazing intensity (e.g., Does nutrient X increase or decrease with increasing grazing intensity? Does it increase at mid-grazing intensity and then decrease?)

Line 240-248: Similar to the previous section, this part, which highlights the impact of grazing on alpha diversity, should be revised to reduce confusion. I recommend referring to the previous comment to revise this section. Additionally, the authors are using several terms interchangeably, such as alpha diversity, species richness, and metrics like the Shannon and Simpson indices, and observed OTUs, which can be confusing since these are all metrics of alpha diversity. I suggest being consistent with the terms and indices used throughout the manuscript.

Line 244-246: This sentence seems incomplete. Which treatments are being compared to NG? Please clarify.

Line 251-256: While the authors do a good job describing the individual effects of grazing and soil depth on beta diversity, the extent to which the patterns observed are consistent across factors remains unclear. For example, are the differences in beta diversity between topsoil and subsoil similar across all grazing treatments? Re-running the PERMANOVA to include the interaction term (grazing x soil depth) could directly address this and provide stronger support for one of the study's premises. I also suggest including a table of the PERMANOVA results in the supplementary materials.

Line 259-261: For consistency between sections, I suggest describing results for both topsoil and subsoil here and elsewhere in the manuscript. One summary sentence when no significant patterns are found could suffice.

Line 262-271:Are the taxa (i.e., genera) discussed here the most abundant in the collection or mainly those that showed significant differences among treatments across the entire dataset? If it's the latter, I suggest applying a correction for multiple comparisons in Duncan’s test. Please clarify this here and in the methods section.

Line 306-308: While the authors do a good job citing relevant studies, this sentence is confusing. The results in Table 1 (Total Nitrogen: TN) show no difference between NG and LG or MG, which doesn't seem to align with the findings of the cited studies (initial increase at light intensity). I suggest revising this for a clearer comparison with previous studies.

Line 322: Add "studies" after "Previous" to improve clarity.

Line 328: Be consistent in your use of "OM" and "SOM."

Line 348-351: This sentence could be revised to improve clarity and better convey the potential reasons why the results found here differ from those in other studies. How have changes in soil environment and plant diversity affected the direction of microbial diversity in those studies, and how does this compare to the current study?

Line 385-386: The sentence “For the topsoil, … changes.” seems to repeat similar information as the previous sentence (“The MRT analysis revealed that the main driving factor regulating fungal community changes was SWC in the topsoil,...”). I suggest revising to avoid redundancy.

Line 389-391: As currently written, it is not clear how the authors' findings compare to the cited study focusing on the growth of aerobic and anaerobic bacteria and soil aeration. I recommend revising to either make the connection between these studies clearer or to cite different studies more directly related to the authors’ findings.

Line 399-400: While the authors' evaluation of taxa-specific correlations with nutrient availability at different depths is an important contribution to the literature, the sentence as currently written is incomplete. Please revise for clarity and completeness.

Table 1: I urge the authors to be consistent in reporting their pairwise statistical comparisons. In this table, some of the numbers for each treatment have letters, while others don't, and some only have lowercase letters. I suggest either using letters for all treatments or clearly explaining what the missing letters and bolding represent.

Figure 1: As above, the authors need to be consistent in reporting their pairwise statistical tests. I suggest either including letters on all box plots or clearly explaining why some box plots are missing letters. Additionally, as currently presented, this figure is hard to follow. I suggest ensuring that the figure is organized so that samples being compared are in the same facet.

Figure 2B: Please ensure that the figure is directly associated with the statistical test being done. For example, Figure 2B shows letters and signs that are easy to read. I also suggest that the authors present these results differently if they want to highlight the results of Duncan's test.

Figure S1: This figure, which highlights some valuable information, seems not to be mentioned in the text. I recommend including a reference to this figure in the manuscript.

Table S1: Please provide more details on the numbers reported. Are these mean values or the number of reads? Clarification is needed

Reviewer 2 ·

Basic reporting

The manuscript needs further polished in language.

Experimental design

Details about the stocking rates are need. For example, the defination of light, moderate, heavy and over grazing.

Validity of the findings

yes

Additional comments

Tu et al. report the results of a long-term grazing experiment in the Inner Mongolia grassland, with focusing on the effects of four grazing intensities on fungal communities at two soil layers (0-20cm and 20-40cm). They found that fungal α-diversity was higher in subsurface soil than in top soil, and that fungal α-diversity decreased with the increase of stocking rate from light to over grazing. They ascribed these differences to differed soil nutrient environments, such as soil organic matter, total nitrogen, soil moisture and concentration of nitrate. These findings are important in that the subsurface soil has a high fungal diversity, which is greatly shaped by soil environments and affected by grazing in grasslands, because the microorganisms in subsoil are not well understood. I only have two concerns about the methods and one suggestion about the Abstract.
First, you should explain why the stocking rate gradient in this study represent light, moderate, heavy and over grazing? How do you define light, moderate, heavy and over grazing? Are the stocking rates (i.e.,1.54 sheep/ha, 1.92 sheep/ha, 2.31 sheep/ha, and 2.69 sheep/ha) are calculated based on growing season or the whole year?

Second, you used one-way ANOVA to compare the difference between treatments under two soil depths. Based on your experiment design, a two-way ANOVA should by adopted because soil depth and grazing treatments have interactions.

Third, the main findings of this study are not well abstracted in the Abstract section. You only repeated the results. In my opinion, the changes in soil environmental factors are not the main findings in this study but only the variables to explain the changes in soil fungi.

Reviewer 3 ·

Basic reporting

In this study, the authors used a range of grazing levels in a desert steppe to determine the effect of livestock grazing on top soil and subsurface soil. The question is an important one and the authors present a paper that has both basic science and management implications, both of which are important. Overall, the figures are clearly made, though I make suggestions below about how to change some of the analysis that, if followed, would result in changing several figures.

Comments on specific lines:
(80, 95) Remove the period before the citations
(82) typo: remove strange dash before “limit”
(160-173) Please provide references for the microplate fluorescence method.
(176-191) Please provide a reference for the CTAB method
(251-256) According to Figure 2B, the dominant phyla are really just Ascomycota, Basidiomycota, Mortierellomycota, and Glomeromycota; the other ones you list here appear to be present at around or below 1% relative abundance according to Figure 2B.
It’s very interesting that you found significantly higher levels of Glomeromycota in heavily grazed plots!
(276-279) Several grammar corrections: should be “were significantly correlated with…” not “correlation with…” (correlated = verb; correlation = noun; for example “A had a correlation with B” is correct as is “A was correlated with B”).
(308-309) It’s strange to read that “the main sources of soil nitrogen are … others;” consider rephrasing to “livestock manure and litter are two main sources of soil nitrogen” or something similar.
(317-325) That’s interesting to see effect of grazing on SWC
(322) Typo: do not capitalize “The”

Experimental design

My only major comment on the experimental design is to ask why the authors treated grazing intensity as discrete categories rather than as continuous variables? Since you know the grazing density in each plot (lines 140-147), you could treat them as continuous and see the shape of any response curves. Also, this would eliminate any potentially artificial distinctions created by the discrete levels you chose to use. For example, is 1.54 sheep/ha versus 1.92 sheep/ha really different enough to justify calling one “low grazing” and the other “medium grazing” intensity? It might be I just don’t know enough about typical grazing pressures in this system to answer that and don't see a specific justification for those levels in the text. I understand these are relative differences within your study, which makes sense. But without clearly articulated reasons for why each of these levels represents *actual* “light,” “moderate,” “heavy,” and “over” grazing as it pertains to actually grazing pressures reflected in this system, the reader has no way of knowing if these are just relative differences within your study or if they represent biologically meaningful differences within the context of typical grazing in this system. You could address this by providing a better justification for your grazing levels or (my recommendation) by treating grazing intensity as a continuous variable and doing a regression-based analysis. This would not only avoid the issues with artificially distinct categories, but would also allow you to see the shape of any response curves, which could provide useful/interesting information.

Finally, treating the data as continuous may impact the statistical power available. Currently, you have n=3 replicates for each of five treatments, leading to fairly low power (though to be fair, I haven't seen a power analysis). You were able to see some significant differences among groups because of the relatively large effect sizes for some measured variables (which shows the big effect of grazing!). If you used a regression-based analysis with continuous data, you may not be able to say as much about differences between specific treatments, but you could more confidently say things about the overall shape of the trend.

Validity of the findings

The authors consolidated the three soil samples from each plot prior to analysis, which would mask any heterogeneity in the landscape. This is not a fatal flaw by any means and combining these technical replicates together is good practice from a statistical standpoint since not doing so would lead to pseudo-replication and a false over-inflation of power. Nevertheless, I believe the discussion would benefit from at least a few sentences discussing the potential impact of grazing on heterogeneity in the landscape and how they may or may not have been able to detect some of those effects given their design. Again, this does not need to be a whole section, just a few sentences would suffice.

Why did you use the 20 cm cutoff for differentiating top soil versus subsoil? Is there a relevant soil horizon there? I don’t think you need a super detailed description, but it would be nice to have some justification for that depth, especially for readers who are not familiar with the soil series in this particular locale.

Additional comments

I struggled to decide whether to choose "accept with minor revisions" or "accept with major revisions." If the authors can clearly and convincingly articulate why they chose the grazing intensity levels they did and why those specific discrete categories represent a priori distinctions that are relevant in this system, then the text would only require relatively minor edits to the writing. Absent that though, I would suggest the authors reanalyze their data using grazing intensity as a continuous (rather than discrete) variable, which, while it may or may not change the primary results, would necessarily change the data structure and therefore all of the major figures and tables in a way that I could warrant an additional review.
Please note however that I think this is a very interesting study and would like to see it published once they make these changes! We need more studies looking at the effects of land use management practices on subsurface microbial communities like this and on the whole, this study does that in a straightforward and informative way!

---

## Round 0.2 · Minor Revisions

The reviewers have provided feedback on your revised manuscript. Please check their comments carefully and revise accordingly. Additionally, I have the following minor comments:

Figure 1 -- the caption should explain the meaning of the lettering above the bars.
Figure 3 -- the caption should explain the meaning of the asterisk(es) on the graph.
Figure 4 -- the caption should explain the meaning of the light and dark blue bars.

Reviewer 1 ·

Basic reporting

The authors have made extensive revisions to the manuscript, significantly improving its quality. The revised version addressed many of the previous comments and suggestions effectively. My remaining suggestions are relatively minor and focus on language refinement, clarity in the presentation of results, and methodological clarifications. Below are some of my specific suggestions:

L35: The term "eutrophic fungi" is confusing and not commonly used. Since "copiotrophic fungi" is used elsewhere in the manuscript, I suggest replacing "eutrophic fungi" with "copiotrophic fungi" for consistency.

L271–273: The sentences "The correlation between fungi and soil environmental factors varied with soil depth" and "Through heatmap analysis, it was found that there had a significant correlation between soil physicochemical properties and soil fungi" are repetitive. Consider combining and rephrasing to say something like "The correlation between fungi and soil environmental factors varied with soil depth, as heatmap analysis revealed significant relationships between soil physicochemical properties and fungi, with distinct patterns observed in the topsoil and subsoil"

Figure 2: Add notes explaining the legend. Specifically, clarify what "1" and "2" represent at the end of each treatment and whether "CK" refers to "NG."
Figure 4: Indicate what A, B, C, and D represent (e.g., topsoil versus subsoil). Are these results combined across all soil depths or separated by depth?

Experimental design

No comment

Validity of the findings

It is still confusing how the define oligotrophic and copiotrophic fungi and the inferences they draw from those. Unless, i am mistaken, copiotrophic fungi are typically fast-growing species that thrive in nutrient-rich environments and able to decompose labile carbon, while oligotrophic fungi are slow-growing, and more adapted to decomposing recalcitrant carbon (Chen et al. 2021, Yang et al. (2023). Hence I will suggest the authors clarify the definitions of oligotrophic and copiotrophic fungi and ensure these definitions align with the conclusions and findings.

Chen, Yongjian, Julia W. Neilson, Priyanka Kushwaha, Raina M. Maier, and Albert Barberán. 2021. “Life‐History Strategies of Soil Microbial Communities in an Arid Ecosystem.” The ISME Journal 15: 649–57. https://doi.org/10.1038/s41396-020-00803-y
Yang Y, Dou YX, Wang BR, Xue ZJ, Wang YQ, An SS, and Chang SX. 2023. Deciphering
628 factors driving soil microbial life-history strategies in restored grasslands. Imeta 2. ARTN e66 10.1002/imt2.66

Additional comments

L57: Replace the comma after "(Dai et al. 2002)" with a period or change "Appropriate" to lowercase.
L58: Add a space between "biomass" and the parentheses.
L59–60: Remove the large space between "limit" and "the ecological."
L71–72: The sentence is unclear. Are you suggesting that increased threshold elemental ratios and microbial biomass could make more organic matter available in soil? Please clarify.
L87–88: Revise to avoid redundant use of "these" to refer to different subjects (e.g., plants, changes, and behaviors).
L 120: " Each site was sampled twice" Does this mean that two separate sample from each soil depth was taken per plots? Or do you mean that in each plot one surface and one subsurface soil samples were taken.
L124: Define what "Sunite" represents or correct if it is a typo.
L133: Please clarify whether the depth division was based on prior observations or studies by the research group. If so, include appropriate references.
L 169 - 171: The location of this sentence is somehow confusing, as taxonomic assignment is generally done after sequencing and some level of denoising. But as currently written is seem like it is done prior library prep? Is that what you did? Also how was the taxonomic assignment of ASVs done?
L183–184: A one-way ANOVA may not fully capture the effects of grazing and soil depth on fungal alpha diversity. Consider using a two-way ANOVA to explore interactions between these factors. If a two-way ANOVA was performed, please report the p-values to support the claim on L227: "The effect of grazing on fungal alpha-diversity was also related to soil depth." Including a supplementary table similar to Table 1 might be helpful.
L193–195: Briefly mention that PERMANOVA and PCoA were performed to analyze beta diversity.
L233: "But not significantly so" seems incomplete or contains a typo. Consider revising for clarity.
L241–242: This sentence is somehow confusing as the PERMANOVA results in Figure 2 show no significance for the interaction factor grazing x soil depth, suggesting that soil depths has very low effect of changes in beta diversity due to grazing and the patterns are the same across the soil depths.
L306: Capitalize "L" in "livestock."
L349–351: The example provided mixes grazing impacts on SOC and intercropping effects on tomatoes and onions, making it unclear how these findings support the authors’ conclusions. Consider rephrasing for better alignment and clarity.

Reviewer 3 ·

Basic reporting

No comment

Experimental design

See comments below.

Validity of the findings

The discussion ends somewhat abruptly without much clear discussion of the bigger picture implications. For example, the last few sentences essentially jump from a nuanced discussion of a change in the relative abundance of certain bacterial phyla at depth directly to a closing statement about soil fertility. Including a paragraph providing a more "zoomed out" contextualization of the study's key findings (beyond that grazing affects soil fertility) would help a lot. What are the implications of how grazing affects soil fertility and how might that point to directions for future research? I don't think this is strictly necessary and the paper could stand alone without it if the authors prefer or if space limits inclusion, but I'd suggest a little more context in the conclusion.

Additional comments

The authors adequately responded to almost all of my comments and I appreciate their efforts. Along with other changes in response to the other reviewers' comments, I think the manuscript is definitely in a stronger position.

My only remaining major feedback is that I still feel the paper needs work contextualizing the grazing intensities that the authors used. Specifically, why did they choose the grazing intensity levels that they did and how do those levels correspond to real-world pastures? I think this is important in order to be able to compare their findings with those of others.

The authors refer to Dai et al. 2022 (CATENA) and Du et al. 2023 (SBB) in their response to reviewers to justify their use of treating grazing intensity as a discrete variable (which I accept), but they still do not adequately contextualize their treatment levels relative to other studies and/or real-world pastures. To reiterate, I don’t have an issue using grazing as a discrete variable per se, but the authors have not yet demonstrated that the levels they chose represent a realistic range of grazing intensities found in this or similar systems or in the absence of that, giving another a priori reason for choosing this range. Is a rate of 2.69 sheep/ha really “over-grazing?” Is 1.92 sheep/ha really “moderate grazing?” If so, relative to what? I just can’t evaluate that based on the present information.

For example, the site used by Dai et al. is roughly 90 km away and also looked at the effect of sheep grazing intensity on soil properties. However, Dai et al.’s study used four grazing intensity levels ranging from 0 sheep/ha (“non-grazed”) to 0.45 sheep/ha (“heavy grazing”). In the current study, the authors use a range from 0 to 2.69 sheep/ha to span their range from “no grazing” to “over grazing.” How is it that the current study defines 1.54 sheep/ha as “light grazing” but Dai et al. refers to 0.45 sheep/ha as “heavy grazing,” especially since the two studies were apparently conducted relatively close to one another in a similar environment with similar sheep? I know that local context matters so this might be appropriate, but that local context is currently missing from the manuscript. If a reader wants to compare the effects of “heavy grazing” on soil properties in different studies, I’m not convinced the current terminology makes sense. Put simply, without knowing more about the system, I have no way of knowing whether 2.69 sheep/ha is actually “over-grazed” (as well as whether their other levels are appropriate either).

Du et al. (2023) actually provide a very good example of how to do this. Du et al. cited native herdsmen who said “the average stocking rate during the last decades was approximately 3 sheep ha-1” and then they included a brief discussion of how some private pastures were grazed much more heavily. They then choose three levels of grazing intensity for their study: 0 sheep/ha (“grazing exclusion”), 3 sheep/ha (“light grazing”), and 9 sheep/ha (“heavy grazing”). They provided reasonable explanations for why they chose the experimental levels they did and they showed how their treatment levels actually correspond to real world grazing intensities. I recommend the authors provide some kind of context like this. Even if it’s anecdotal, it would help to have some context.

I apologize if it seems like I am nitpicking a detailed point, but without some effort to contextualize their treatment levels in a real-world context, it’s hard to evaluate the implications of this work in other systems. This is especially important when it comes to terms like “over-grazing,” since this means different things in different places and can have important implications. Fortunately, if the authors can contextualize the grazing intensities they used, it will help readers delineate these better. Just a little more context would go a long way!

---

## Round 0.3 · accepted · Accept

Regarding the concluding sentence of the manuscript, I suggest replacing "provides empirical support for solving the impact. of grazing.." with "provides empirical evidence of the impact of grazing..."